# Dramatic Decrease of Vitamin K2 Subtype Menaquinone-7 in COVID-19 Patients

**DOI:** 10.3390/antiox11071235

**Published:** 2022-06-24

**Authors:** Harald Mangge, Florian Prueller, Christine Dawczynski, Pero Curcic, Zdenka Sloup, Magdalena Holter, Markus Herrmann, Andreas Meinitzer

**Affiliations:** 1Clinical Institute of Medical and Chemical Laboratory Diagnostics, Medical University of Graz, 8036 Graz, Austria; florian.prueller@medunigraz.at (F.P.); p.curcic@medunigraz.at (P.C.); zdenka.sloup@medunigraz.at (Z.S.); markus.herrmann@medunigraz.at (M.H.); andreas.meinitzer@medunigraz.at (A.M.); 2Institute of Nutritional Science, Friedrich Schiller University Jena, 07743 Jena, Germany; christine.dawczynski@uni-jena.de; 3Institute of Medical Computer Sciences, Statistics and Documentation, Medical University of Graz, 8036 Graz, Austria; magdalena.holter@medunigraz.at

**Keywords:** COVID-19 pneumonia, non-COVID-19 pneumonia, different subtypes of vitamin K

## Abstract

**(1) Background**: Vitamin K (VK) is a fat-soluble compound with a common chemical structure, a 2-methyl-1,4-naphthoquinone ring, and a variable aliphatic side-chain. VK is involved in the synthesis of blood-clotting proteins, bone stability, anti-oxidative, and immune inflammatory-modulatory functions. Vitamin K also activates protein S, which acts as an antioxidant and anti-inflammatory. The fact that cytokine overproduction, oxidative stress, and disturbed microcirculation by thrombogenicity play a central role in severe COVID-19 prompted us to analyze this vitamin. **(2) Methods**: We analyzed by a validated liquid-chromatography tandem mass-spectrometry method serum vitamin K1, MK4, MK7, and VK epoxide levels in 104 healthy controls, 77 patients with non-COVID-19 pneumonia, and 135 hospitalized COVID-19 patients with potentially fatal outcomes admitted to our University Hospital between April and November 2020. We included the quotient between VK and triglyceride (TG, nmol/mmol/L) values in the analyses with respect to the TG transporter function for all VK subtypes. Additionally, we assessed anthropometric, routine laboratory, and clinical data from the laboratory and hospital information systems. **(3) Results**: The COVID-19 patients had significantly lower MK7 levels than non-COVID-19 pneumonia patients and healthy controls. COVID-19 and non-COVID-19 pneumonia patients had significantly lower vitamin K1 and significantly higher MK4 compared to healthy controls, but did not differ significantly from each other. Between COVID-19 non-survivors (*n* = 30) and survivors (*n* = 105) no significant differences were seen in all vitamin K subtypes, despite the fact that non-survivors had higher peak concentrations of IL-6, CRP, d-dimer, and higher oxygen needs, respectively. **(4) Conclusions**: The present data identified significantly decreased vitamin K1, K2 (MK7), and increased MK4 levels in patients with COVID-19 compared to healthy controls. Vitamin K2 (MK7) was lowest in COVID-19 patients irrespective of potentially fatal courses, indicating consumption of this VK subtype by COVID-19 immanent effects, most probably inflammatory and oxidative stress factors.

## 1. Introduction

Coronavirus disease (COVID-19) is a global health problem. As of 13 June 2022, there have been 532,887,351 confirmed cases worldwide, including 6,307,021 deaths (https://covid19.who.int/, accessed on 13 June 2020).

The world continues to suffer from successive waves of the pandemic, fueled by the emergence of viral variants [1]. Moreover, persistent, prolonged, and often debilitating sequelae are increasingly recognized in convalescent individuals, called “post-COVID-19 syndrome” [1]. The underlying pathophysiological mechanisms are so far poorly understood [1].

The main pathologic feature of severe COVID-19 is an excessive inflammation with a central role of interleukin (IL)-6, which is consistently upregulated [2,3,4,5,6,7]. In SARS-CoV-2 pneumonia, IL-6 production originates mainly from the pulmonary compartment and promotes consecutive immune cell infiltration [8,9] and severe inflammation with tissue destructive components, including the release of matrix metalloproteinases (MMP) by macrophages. This process alters lung and vascular structures [10] and is associated with extensive pulmonary endotheliitis [11], and ACE2 protein expression [12] which leads to hypercoagulability with arterial, microvascular, and venous thrombosis [10]. A multi-organ involvement is usually a characteristic autopsy feature [11]. Vaccination is safe and considered the best approach to control the pandemic. As of 13 June 2022, reported to WHO, a total of 11,854,673,610 vaccine doses have been administered (https://covid19.who.int/, accessed on 13 June 2020). Nevertheless, too many people refuse the vaccination and show severe courses of COVID-19 with potentially fatal lung involvement. Thus, the search for potential therapeutic agents is ongoing. Because of discussed antioxidant, anti-inflammatory, and balancing clotting effects, vitamin D and K came into the focus of COVID-19 research [10]. Concerning vitamin D, recent studies show contradictory results [13,14], and we could not find any correlation between this vitamin and its metabolites for clinical outcome in our COVID-19 cohort [15]. This cohort was set up at the Medical University of Graz to find new risk biomarkers for severe courses of COVID-19. Stimulated by our well-developed analytical LC-MS/MS expertise for vitamin subtype measurements and other studies reporting a significant involvement of vitamin K in COVID-19 pathology [16,17,18], we decided to analyze a spectrum of the most important vitamin K subtypes in frozen samples of our COVID-19 cohort. As controls, we used non-COVID-19 pneumonia patients and healthy controls, all age and sex-matched.

Vitamin K is a cofactor in the activation of vitamin K-dependent coagulation factors (CF) [19] which belong to the family of vitamin K-dependent proteins (VKDP). The carboxylation of glutamate (Glu) residues is a central process for the activation of VKDPs. In the case of CFs, with the support of calcium, vitamin K enables high-affinity binding of CFs to negatively charged phospholipid membrane, which maintains fine clotting hemostasis including pro- and anticoagulant protein factors [19].

Recently, other members of the VKDPs have become the focus of interest. For example, vitamin K-dependent activation of matrix Gla protein protects soft tissues by blocking mineralization and degradation. This dampens inflammation-induced vascular and pulmonary tissue damage [16] and is protective against atherosclerosis [19]. Vitamin K may also suppress IL-6 production, both indirectly through its activation of the immune inhibitory proteins growth-arrest-specific gene 6 (Gas6) [20] and protein S [21], or directly by inhibiting phosphorylation of IKKa/b that is required for activation of nuclear factor (NF)kB [22,23,24,25].

Vitamin K1 and especially K2 (MK4, MK7) act as antioxidants. A paralogue enzyme of vitamin K oxidoreductase 1 (VKORC1), named VKORC1-like one (VKORC1L1), is expressed in many tissues. VKORC1L1 mediates vitamin K-dependent intracellular antioxidant effects including lipid peroxidation in the human cellular membrane [26] with a 10- to 100-fold higher activity than radical scavengers, such as alpha-tocopherol and ubiquinone [27,28]. Coumarins like warfarin break this anti-oxidative activity. Both vitamin K1 and K2 prevent oxidative stress in neuronal cells and primary oligodendrocytes through the inhibition of 12-lipoxygenase [29,30]. Vitamin K also facilitates ATP generation and is involved in the rescue of mitochondrial dysfunction [31].

These functions prompted us to analyze vitamin K1, MK4, MK7, and vitamin K epoxide levels in 135 hospitalized COVID-19 ALDACOV patients compared to a control group of 77 patients with non-COVID-19 pneumonia and 104 healthy controls with a common western lifestyle.

## 2. Material and Methods

### 2.1. Study Design

We established a biobank (Alpe_Adria_Coronavirus_Cohort, ALDACOV) by collecting leftovers of blood samples from patients suffering from COVID-19 whenever sent to the central laboratory of our university hospital between April and December 2020 (Ethical vote 32-475 ex 19/20). After the completion of all routine laboratory testing, residual material was stored at −80 °C until batched analysis. Several biomarker studies were recently published [3,15]. Herein, we measured the serum concentrations of vitamin K1, MK4, MK7 and vitamin K epoxide by a validated LC-MS/MS [32] method in leftover serum samples from 148 COVID-19 patients admitted to the University Hospital of the Medical University of Graz between April and November 2020. The following biomarkers were available from the ALDACOV data repository: plasma concentrations of kynurenine (KYN); tryptophan (TRP); ferritin; IL-6; C-reactive protein (CRP); creatinine; N-terminal-pro hormone B-type pro-natriuretic peptide (NTproBNP); troponin T (TnT); D-Dimer; prothrombin time (PT); activated partial thromboplastin time (aPTT); antithrombin (AT); protein C; free protein S; factor XIII. Basic clinical characteristics and anteceding diseases including cardiovascular, oncologic, renal, hypertension, pulmonary, and metabolic (diabetes and obesity) were recorded in the database. Anthropometric and clinical data, as well as outcome data, were obtained from the laboratory and hospital information systems. The primary outcome was death within 90 days after admission. Respiratory support with oxygen was used as the secondary endpoint. The institutional ethics committee of the Medical University of Graz (EK 32-475 ex 19/20) approved this work. Control groups were 77 patients with non-COVID-19 pneumonia from our university hospital, and 105 healthy persons provided from the KoALA study, Jena, Germany (NCT03558776). The KoALA participants were recruited in 2018 and are characterized by following a traditional western diet and normal to moderate elevated plasma triglyceride concentrations (1.26 ± 0.57 mmol/l).

### 2.2. Laboratory Analysis

The diagnosis of COVID-19 was confirmed in all patients with viral reverse transcriptase PCR using the Xpert^®^ Xpress SARS-CoV-2 (singleplex) cartridge and device (GeneXpert, Cepheid GmbH, 47807 Krefeld, Germany), as previously described [33]. Interleukin-6 (IL-6), ferritin, NTproBNP, cTnT, and C-reactive protein (CRP) were measured with commercial immunoassays on a COBAS 8000 analyzer (Roche Diagnostics, Rotkreuz, Switzerland). We measured vitamin K by a newly developed LC-MS/MS method [32]. The overall precision of all analytes (combination of intra and inter-assay CVs) ranged between 4.8% and 17.7%. D-Dimer, prothrombin time, activated partial thromboplastin time (aPTT), antithrombin (AT), protein C, free protein S, and factor XIII were analyzed on a Siemens Atellica COAG-360 analyzer (Siemens Healthineers, Marburg, Germany).

### 2.3. Data Analysis

For descriptive statistics, the median, 25th, and 75th percentiles were determined for continuous, not normally distributed variables. Categorical outcome variables were displayed as absolute and relative frequencies, and differences were assessed either by a chi-square test or by Fisher’s exact test. Continuous outcomes for independent samples were analyzed by Mann–Whitney U and Kruskal–Wallis tests. Due to multiple testing, we used a Bonferroni correction: *p*-values < 0.004 were considered significant. Statistical analyses were performed using SPSS statistical software (version 26.0; IBM Corp., Armonk, NY, USA) and SAS software (version 9.4; SAS Institute, Inc., Cary, NC, USA).

## 3. Results

Table 1 shows the baseline characteristics of the COVID-19 patients. People who survived COVID-19 were younger on average (*p* < 0.001, Mann–Whitney U test) and had an increased oxygen need. No significant difference was found in survival regarding sex (*p* = 0.855, chi-square test). At the time of SARS-CoV2 diagnosis, COVID-19 non-survivors had significantly higher CRP, IL-6, D-Dimer, and antithrombin levels compared to the survivors. Vitamin K subtypes did not differ between these groups, neither absolutely nor after normalization by TG levels (Table 1).

Table 2 and Figure 1 depict that the COVID-19 patients had markedly decreased MK7 levels compared to patients with pneumonia and healthy controls. Correction by TGs strengthened the effect. Both COVID-19 and non-COVID-19 pneumonia patients had significantly lower vitamin K1, and significantly, higher MK4 compared to healthy controls but did not differ significantly from each other with respect to these vitamin K subtypes.

## 4. Discussion

The central observation of this study is the very low serum MK7 in patients with COVID-19 pneumonia compared to non-COVID-19 pneumonia and healthy controls. Together with MK4, MK7 forms vitamin K2. Vitamin K2 has a longer half-life in the circulation than vitamin K1 [25,34,35]. While vitamin K1 remains in the liver with its well-established role in coagulation, vitamin K2 redistributes to the circulation and extrahepatic tissues. Vitamin K2 involves in bone metabolism [36], cardiovascular disease (CVD), chronic kidney disease, and cancer [37]. Moreover, it plays a role in liver disease, neurological disease, obesity, and immune response.

Two vitamin K2 homologues, menaquinone-4 (MK-4) and menaquinone-7 (MK-7) exist. It is still open to what extent these fractions contribute to the complex vitamin K2 functions. A recent comparison of MK4 and MK7 bioavailability in healthy women showed that MK-4 present in food does not contribute to vitamin K status [38]. In contrast, nutritional MK-7 significantly increases serum MK-7 levels and therefore may be of particular importance for the vitamin K2 actions in the extrahepatic body compartments [38]. In the liver, phylloquinone is converted to menadione, a metabolic intermediate that is converted subsequently by cellular alkylation to MK-4 but not to other menaquinones. The enzyme UbiA prenyltransferase domain-containing protein 1 (UBIAD1) has been identified in humans and catalyzes the initial side-chain cleavage of phylloquinone to release menadione. By prenylation of menadione, MK-4 is formed [39]. This conversion appears also to occur in other organs, which is indicated by the fact that MK-4 levels exceed vitamin K1 levels in tissues other than the liver [40].

The question remains why MK7 levels are so specifically low in COVID-19 pneumonia. COVID-19 pneumonia-induced extrahepatic vitamin K depletion may play a role [17]. Matrix Gla protein (MGP) protects against pulmonary and vascular elastic fiber damage. This protective effect needs vitamin K-mediated carboxylation [41]. In our COVID-19 pneumonia patients, the low MK7 may indicate an extrahepatic wastage of this vitamin K subtype. Interestingly, as shown in a recent study COVID-19 patients with severe clinical course had normal levels of hepatic pro-coagulant factor II (FII), indicating normal vitamin K function in the liver [17]. In contrast to this, these patients suffered from an extrahepatic vitamin K insufficiency/deficiency associated with severely increased inactive MGP, indicating a failed vitamin K mediated carboxylation of MGP. This inactive MGP contributes to pulmonary and vascular elastic fiber damage and leads to decreased protein S synthesis because of impaired endothelial cells. In contrast to FII, which is of solely hepatic origin, protein S is also extra-hepatically synthesized in endothelial cells [42], which explains the deficit. Thus, this situation can paradoxically lead to enhanced thrombogenicity in a state of low vitamin K [42].

Furthermore, MGP is well-known as a calcification inhibitor in arterial walls [43]. Notably, MGP’s role in the pulmonary compartment seems to be comparable [44]. Elastic fibers are essential matrix components in the lungs and have a high calcium affinity [45]. Degradation and mineralization of elastic fibers are interrelated processes [46,47]. Matrix metalloproteinases (MMPs) synthesis increases in parallel with elastic fiber calcification [48], and partially degraded elastic fibers become prone to mineralization [45]. Recent data show that a subset of MMP-producing macrophages increases in severe SARS-CoV-2 pneumonia [49]. Thus, in COVID-19 pneumonia, both MK7 deficiency and hyper-inflammation potentiated elastic fiber deterioration may be linked through a series of sequential pathologic steps.

Individuals with severe SARS-CoV-2 infections often have comorbidities that are associated with reduced vitamin K status, such as hypertension, diabetes, and cardiovascular diseases [43,50]. In this case, therapy with vitamin K antagonists like Warfarin carries the risk of vascular calcification and may have contributed to the low MK7 effect seen in our COVID-19 pneumonia patients. However, the percentage of non-COVID-19 pneumonia patients with vitamin K antagonist therapy was low (4.4%, verified by vitamin K epoxide measurements) and did not differ significantly from non-COVID-19 pneumonia patients (3.9%). This argues against such an influence.

The body uses vitamin K very efficiently, and storage capacity is low [51]. There are reasons to suspect that there is increased utilization of vitamin K for carboxylation of pulmonary MGP and coagulant factors during COVID-19. As stated above, data suggest that COVID-19 pneumonia-induced extrahepatic vitamin K depletion leading to accelerated elastic fiber damage and thrombosis risk due to impaired vitamin K-dependent activation of MGP and endothelial protein S, respectively [17]. As MK7 is lipophilic, the pathologic lung involvement may consume it in COVID-19 pneumonia in a particularly strong way to counteract this mechanism, a fact that may explain the observed low MK7 levels seen in COVID-19 pneumonia.

Another attribute clearly related to MK7 comprises immunomodulatory functions by modulation of TNF-α, IL-1α, and IL-1β expression [52,53,54,55]. MK7 may also decrease proliferation of T-cells suggesting a novel immunosuppressive role [56,57]. Thus, the low MK7 seen in our patients may represent a counter-regulative reaction to break the overwhelming inflammatory reaction seen in the lungs of COVID-19 patients. Additionally, the antioxidant functions of MK7 may play a part in the process in this context [28].

The question remains why non-COVID-19 pneumonia patients did not show these low MK7 levels. Considering frequently administered antibiotic therapy in these patients, one may think that antibiotic reduced intestinal bacteria diversity is responsible for the decreased MK7. However, in our study, both COVID-19 pneumonia and non-COVID-19 pneumonia patients received antibiotic therapy. Thus, the question of why low MK7 is associated specifically with COVID-19 pneumonia is still open. Most probably, the COVID-19-specific pulmonary pathologic surrounding, and the intense inflammation paved the way for this wasting phenomenon.

Summarized, our results suggest a disturbed vitamin K metabolism as a potential missing link between COVID-19-induced lung damage and thromboembolism. Nevertheless, the therapeutic potential of vitamin K supplementation in COVID-19 remains largely unexplored [16]. Clinical intervention studies are required to clarify this [58]. Given the ongoing COVID-19 pandemic and the extremely high prevalence of pronounced vitamin K deficiency in severe COVID-19 patients, such studies are justified in a strong way [58]. The low toxicity of vitamin K in humans may ease such attempts. The possible in vivo benefit of a joint administration with vitamin D shown to be protective in other infectious diseases is encouraged to be investigated [59].

The limitations of this study are as follows. Deeper analyses to satisfactorily explain the pathophysiological mechanisms of COVID-19 specific MK-7 depletion are missing. These analyses, will include, amongst others the measurement of proteins induced by vitamin K absence (PIVKAs) and dephosphorylated-undercarboxylated matrix Gla protein (dp-ucMGP) and undercarboxylated osteocalcin levels to better understand the status of intra and extrahepatic vitamin K dependent proteins (VKDPs) in the COVID-19 patients. Another limitation is that we were not able to determine the adequateness of nutritional vitamin K2 support in the COVID-19 patients. In this context, the ineffectiveness of current food frequency questionnaires is noteworthy [60]. Furthermore, the investigated COVID-19 group was too small to effectively test the clinical effects of a vitamin K2 substitution treatment.

## 5. Conclusions

The present results show abnormalities in the concentrations of vitamin K subtypes in patients with COVID-19 pneumonia. In particular, vitamin MK7 is severely low compared to non-COVID-19 pneumonia and healthy controls, indicating high expenditure in extrahepatic tissues, especially the lung. Larger studies should confirm our results and clear up the cause of the abnormal vitamin K profile. The link between the COVID-19 specific immune response, oxidative stress, lung damage, and thrombogenicity will be a promising candidate to improve the understanding of the complex extrahepatic vitamin K functions. Vitamin K2 supplementation may be a cheap and effective preventive approach to act against severe courses of COVID-19. 

## Figures and Tables

**Figure 1 antioxidants-11-01235-f001:**
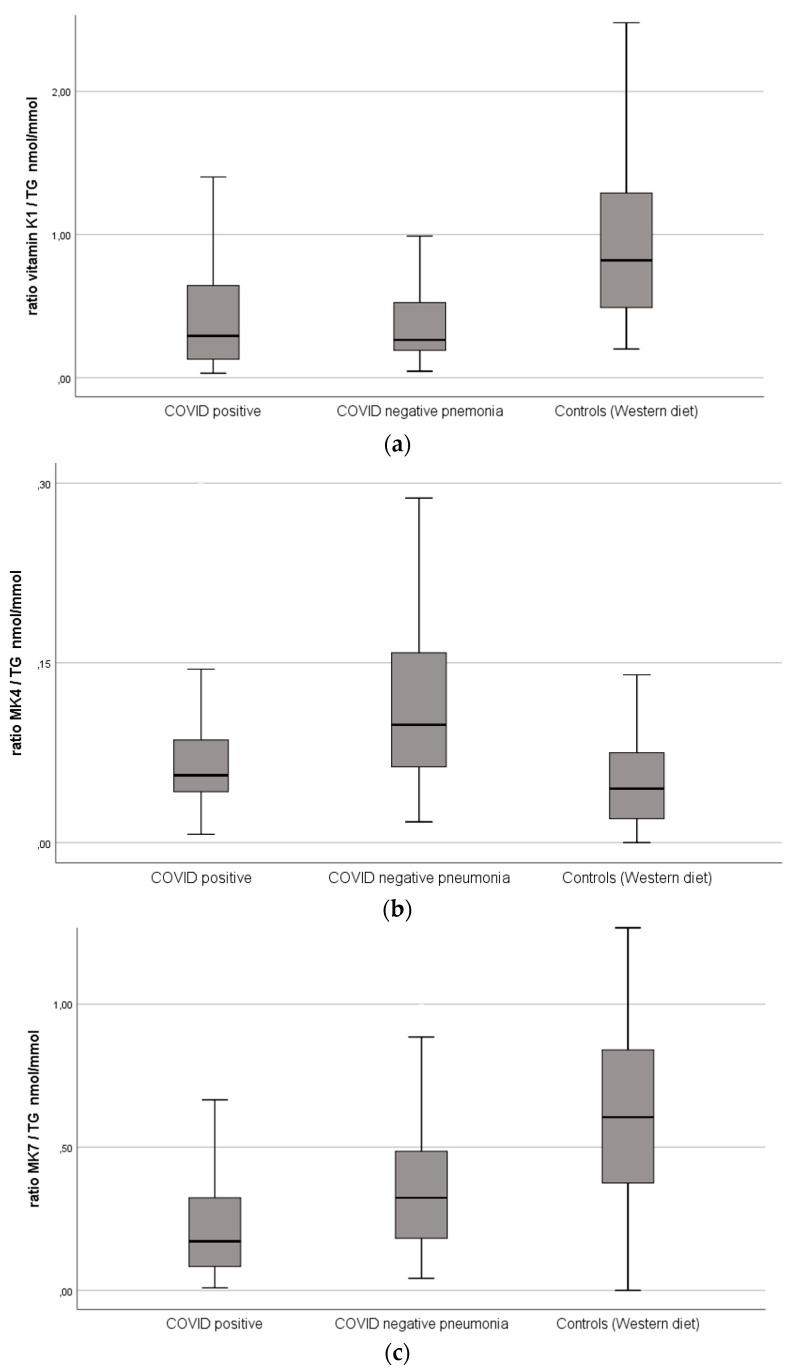
(**a**) Vitamin K1 serum levels normalized by triglyceride levels within the groups. COVID-19 positive, *n* = 135, COVID-19 negative pneumonia, *n* = 77, controls (Western diet), *n* = 104. (**b**) Vitamin MK4 serum levels normalized by triglyceride levels within the groups. (**c**) Vitamin MK7 serum levels normalized by triglyceride levels within the groups.

**Table 1 antioxidants-11-01235-t001:** Baseline characteristics of COVID-19 patients.

	Exitus (*n* = 30)	Recovery (*n* = 105)	*p*-Value
Sex female	15 (50.0%)	50 (47.6%)	
Oxygen therapy	28 (90.3%)	60 (51.7%)	
Age (years)	79.5 (67.7, 87.2)	58.0 (43.5, 78.0)	<0.001
CRP mg/dL	93.1 (33.8, 161.8)	22.6 (4.5, 75.2)	<0.001
IL-6 pg/mL	76.8 (30.1, 192.0)	19.6 (10.3, 63.4)	<0.001
D-Dimer high sensitive µg/L	1820 (659, 3799)	550 (371, 1264)	0.026
Antithrombin %	90 (79, 101)	96 (86, 105)	0.014
Factor XIII %	96.5 (76.5, 130.5)	100 (86, 118)	0.851
Protein C %	100 (68, 109)	94 (80, 109)	0.531
Free protein S %	72 (57, 90)	69 (54, 85)	0.997
Vitamin K1, nmol/L	0.31 (0.17, 0.75)	0.37 (0.14, 0,86)	0.952
Vitamin K1/TG, nmol/mmol	0.18 (0.14, 0.76)	0.31 (0.13, 0.67)	0.576
Vitamin MK4, nmol/L	0.09 (0.07, 0.13)	0.08 (0.06, 0.13)	0.446
Vitamin MK4/TG, nmol/mmol	0.05 (0.04, 0.06)	0.06 (0.04, 0.09)	0.273
Vitamin MK7, nmol/L	0.19 (0.12, 0.44)	0.26 (0.13, 0.51)	0.537
Vitamin MK7/TG, nmol/mmol	0.12 (0.08, 0.28)	0.18 (0.09, 0.37)	0.355

Data are presented in medians and 25th and 75th percentiles or as absolute and relative frequencies. TG = Triglycerides. Differences analyzed by Mann–Whitney U test.

**Table 2 antioxidants-11-01235-t002:** Anthropometric and biochemical characteristics of the study groups.

	Group 1	Group 2	Group 3	*p*-Value	*p*-Value	*p*-Value
	COVID-19 pneumonia *n* = 135	non-COVID-19 pneumonia *n* = 77	Healthy controls *n* = 104	group 1 versus group 2	group 1 versus group 3	group 2 versus group 3
**Basic characteristics**						
Age, years	63.0 (52.0–80.0)	61.0 (47.0–81.0)	62.0 (53.5–72.0)	0.7184		
Sex, female	65 (48.2%)	44 (57.1%)	68 (65.4%)			
**Vitamin K subtypes**, first visit						
Vitamin K1, nmol/L	0.32 (0.16–0.85)	0.37 (0.23–0.73)	0.89 (0.58–1.46)	0.378	<0.001	<0.001
Vitamin K1/TG, nmol/mmol	0.29 (0.13–0.65)	0.26 (0.19–0.52)	0.82 (0.49–1.31)	0.376	<0.001	<0.001
Vitamin K2 (MK4), nmol/L	0.08 (0.06–0.13)	0.14 (0.09–0.22)	0.05 (0.03–0.09)	*<0.001*	*<0.001*	*<0.001*
Vitamin K2 (MK4)/TG, nmol/mmol	0.06 (0.04–0.09)	0.10 (0.06–0.16)	0.05 (0.02–0.08)	*<0.001*	*<0.001*	*<0.001*
Vitamin K2 (MK7), nmol/L	0.24 (0.13–0.5)	0.38 (0.21–0.67)	0.64 (0.47–0.85)	0.004	<0.001	<0.001
Vitamin K2 (MK7)/TG, nmol/mmol	0.17 (0.08–0.3)	0.32 (0.18–0.49)	0.61 (0.38–0.84)	0.001	<0.001	<0.001

Data are presented in medians and 25th and 75th percentiles or as absolute and relative frequencies. TG = Triglycerides. *p* values in italic letters indicate increased levels of vitamin K subtypes. Differences analyzed by Kruskal–Wallis test, overall comparison *p* < 0.001 for all tested vitamin K parameters.

## Data Availability

Data are contained within the article.

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
