# Peer review of "Dramatic Decrease of Vitamin K2 Subtype Menaquinone-7 in COVID-19 Patients"

_antioxidants, 2022, doi:10.3390/antiox11071235_

Round 1

Reviewer 1 Report

I read with great interest the paper. I find it well write and with good idea research

Below my suggestions:

1. Introduction: updata data on SARS CoV2 wordwilde

2. Method and results: are very clear and table presented well the paper

3. Discussion: add the role of Vitamin in other infectious diseases not only SARS CoV2 (ex Papagni, R. (2022). Impact of Vitamin D in Prophylaxis and Treatment in Tuberculosis Patients.

4. Conclusion: give some proposal that came from your paper

5. Add limitation section

Author Response

I read with great interest the paper. I find it well write and with good idea research
Re.: Thank you.

Below my suggestions:

  1. Introduction: updata data on SARS CoV2 wordwilde

Re.: We updated the introduction accordingly. Please, refer on lines 36-42, updated references 6, 7 and lines 49-54.

  1. Method and results: are very clear and table presented well the paper

Re.: Thank you.

  1. Discussion: add the role of Vitamin in other infectious diseases not only SARS CoV2 (ex Papagni, R. (2022). Impact of Vitamin D in Prophylaxis and Treatment in Tuberculosis Patients.

Re.: Done (line 259-261).

  1. Conclusion: give some proposal that came from your paper

Re.: Done (line 276-278).

  1. Add limitation section

Re.: Done (line 263-268).

Reviewer 2 Report

Authors should add values of PIVKA (proteins induced by vitamin K absence)

authors should better describe potential clinical effects of their findings in particular in daily clinical practice (regarding outcomes of thrombosis or bleedings)

Author Response

Authors should add values of PIVKA (proteins induced by vitamin K absence)

Re.: This is a very interesting point. Indeed, analysis of PIVKAs would give additional information about defective carboxylation of factors II, VII, IX, X and proteins C as well as S. Unfortunately, we have not enough resources to perform these measurements at the moment. Nevertheless, we comment on the value of this additional analysis in the discussion section by a newly introduced limitations section (line 263-268). Furthermore, the analysis of undercarboxylated osteocalcin levels may help to understand the extra-hepatic effects of the COVID-19 specific vitamin K2 (MK7) deficiency in a better way (line 266-268).

authors should better describe potential clinical effects of their findings in particular in daily clinical practice (regarding outcomes of thrombosis or bleedings)

Re.: We added a section dealing with this important point (lines 254-262).

Round 2

Reviewer 2 Report

Raised issues were focused by authors without substantial furiò al data . So I’m order to underline their work I suggest to better define the study limitation paragraph. 

Author Response

Raised issues were focused by authors without substantial furiò al data . So I’m order to underline their work I suggest to better define the study limitation paragraph. 

Re.: We overhauled the limitation paragraph. A better definition of the study limitation paragraph is included in the second revision of the manuscript.